# Isolation of Neoantigen-Specific Human T Cell Receptors from Different Human and Murine Repertoires

**DOI:** 10.3390/cancers14071842

**Published:** 2022-04-06

**Authors:** Corinna Grunert, Gerald Willimsky, Caroline Anna Peuker, Simone Rhein, Leo Hansmann, Thomas Blankenstein, Eric Blanc, Dieter Beule, Ulrich Keller, Antonio Pezzutto, Antonia Busse

**Affiliations:** 1Department of Hematology, Oncology and Cancer Immunology, Campus Benjamin Franklin, Charité–Universitätsmedizin Berlin, 12203 Berlin, Germany; caroline-anna.peuker@charite.de (C.A.P.); ulrich.keller@charite.de (U.K.); antonio.pezzutto@charite.de (A.P.); 2Max Delbrück Center for Molecular Medicine in the Helmholtz Association, 13092 Berlin, Germany; simone.rhein@mdc-berlin.de (S.R.); tblanke@mdc-berlin.de (T.B.); 3Institute of Immunology, Campus Buch, Charité–Universitätsmedizin Berlin, 13092 Berlin, Germany; gerald.willimsky@charite.de; 4German Cancer Research Center, 69120 Heidelberg, Germany; 5German Cancer Consortium (DKTK), Partner Site Berlin, CCCC (Campus Mitte), 10117 Berlin, Germany; leo.hansmann@charite.de; 6Department of Hematology, Oncology and Cancer Immunology, Campus Virchow-Klinikum, Charité–Universitätsmedizin Berlin, 13353 Berlin, Germany; 7Berlin Institute of Health at Charité, 10178 Berlin, Germany; 8Core Unit Bioinformatics, Berlin Institute of Health at Charité, 10117 Berlin, Germany; eric.blanc@bih-charite.de (E.B.); dieter.beule@bihealth.de (D.B.)

**Keywords:** neoantigens, T cell receptor (TCR) therapy, tumor-specific TCR, antigen-specific T cell, T cell receptor repertoire

## Abstract

**Simple Summary:**

T cell-based immunotherapy has achieved remarkable clinical responses in patients with cancer. Neoepitope-specific T cells can specifically recognize mutated tumor cells and have led to tumor regression in mouse models and clinical studies. However, isolating neoepitope-specific T cell receptors (TCRs) from the patients’ own repertoire has shown limited success. Sourcing T cell repertoires, other than the patients’ own, has certain advantages: the availability of larger amounts of blood from healthy donors, circumventing tumor-related immunosuppression in patients, and including different donors to broaden the pool of specific T cells. Here, for the first time, a side-by-side comparison of three different TCR donor repertoires, including patients and HLA-matched allogenic healthy human repertoires, as well as repertoires of transgenic mice, is performed. Our results support recent studies that using not only healthy donor T cell repertoires, but also transgenic mice might be a viable strategy for isolating TCRs with known specificity directed against neoantigens for adoptive T cell therapy.

**Abstract:**

(1) Background: Mutation-specific T cell receptor (TCR)-based adoptive T cell therapy represents a truly tumor-specific immunotherapeutic strategy. However, isolating neoepitope-specific TCRs remains a challenge. (2) Methods: We investigated, side by side, different TCR repertoires—patients’ peripheral lymphocytes (PBLs) and tumor-infiltrating lymphocytes (TILs), PBLs of healthy donors, and a humanized mouse model—to isolate neoepitope-specific TCRs against eight neoepitope candidates from a colon cancer and an ovarian cancer patient. Neoepitope candidates were used to stimulate T cells from different repertoires in vitro to generate neoepitope-specific T cells and isolate the specific TCRs. (3) Results: We isolated six TCRs from healthy donors, directed against four neoepitope candidates and one TCR from the murine T cell repertoire. Endogenous processing of one neoepitope, for which we isolated one TCR from both human and mouse-derived repertoires, could be shown. No neoepitope-specific TCR could be generated from the patients’ own repertoire. (4) Conclusion: Our data indicate that successful isolation of neoepitope-specific TCRs depends on various factors such as the heathy donor’s TCR repertoire or the presence of a tumor microenvironment allowing neoepitope-specific immune responses of the host. We show the advantage and feasibility of using healthy donor repertoires and humanized mouse TCR repertoires to generate mutation-specific TCRs with different specificities, especially in a setting when the availability of patient material is limited.

## 1. Introduction

Remarkable clinical responses in patients with cancer have been achieved by T cell-based immunotherapy, especially in those with high mutational burden. Subsequent events of single point mutations may be sufficient to drive an oncogenic transformation in certain cancers [1]. Non-synonymous, somatic mutations can further accumulate in tumor cells over time, potentially resulting in the generation of targetable neoepitopes. Increasing evidence in mouse models and clinical settings has shown that T cells specific to neoepitopes can lead to tumor regression after receiving adoptive T cell therapy (ATT) [2,3,4,5], or immune checkpoint inhibitor therapy [6]. Neoantigens have the advantage of being truly tumor-specific, and thus are not associated with on-target toxicity. Moreover, by definition, they should be unable to have induced central tolerance. Redirecting patients’ peripheral T cells by transferring mutation-specific TCRs, targeting a defined tumor-specific antigen, has the potential to become one of the most effective immunotherapeutic approaches. Yet, in addition to the identification of tumor-specific neoepitopes, this strategy requires the isolation of high-affinity neoepitope-specific TCRs. So far, successful isolation of neoepitope-specific T cells has only been achieved for a minority of potential neoepitopes. Although neoepitope-specific T cells can be found among TILs [7,8], they are not easy to isolate and expand [9]. Moreover, they do not necessarily contain the entire repertoire of possible neoantigen-specific T cells [10,11]. Notably, in epithelium-derived tumors most TILs are not tumor-specific but rather virus-specific and incapable of recognizing cognate tumors. Only about 10% of intratumoral CD8+ T cells were reported to recognize autologous tumor cells, and only a minority of reported somatic mutations induce T cell immune responses in epithelial cancers [12,13]. Even target antigens of predominantly TIL clones do not appear to be restricted to tumor tissue [14,15].

Often, the amount and availability of patient tumor material for isolating TILs is limited. In contrast, peripheral blood (PB) of cancer patients is easily accessible. The presence of a neoepitope-specific immune response can be detected in the memory T cell population provided that immunogenic neoepitopes are presented by the tumor and have resulted in successful T cell priming [16]. Furthermore, several studies have shown that PB from patients can be used to prime de novo neoantigen-specific T cells, presenting a useful source for the production of therapeutic T cell products [7]. Peripheral blood might contain a more diverse T cell repertoire in the naive T cell population as compared to TILs. However, so far T cells specific to predicted neoantigens could be isolated from PB only in a minority of cancer patients [11]. The failure to successfully identify tumor-specific T cells and isolate the specific TCRs, from both TILs and PB, could be due to tumor-related immunosuppression impeding the ability to build a potent immune response against neoepitopes during T cell priming at the tumor site. This hypothesis is supported by the observation of Strønen and colleagues, who successfully isolated neoantigen-specific T cell clones from PB of healthy donors, which were not detected in the patients’ TIL [10]. Moreover, we and others have shown that an independent, non-tolerized T cell repertoire of an HLA-compatible healthy donor is a suitable alternative source for the isolation of neoantigen-specific T cells [10,17,18,19]. Although the risk of cross- and alloreactivity is increased when using donor TCR repertoires compared to the patient’s own repertoire, this strategy has certain advantages: larger amounts of blood can be obtained from healthy donors than from most cancer patients, and the difficulty to isolate and expand cells from a presumably very low precursor frequency can be circumvented. 

Using HLA-A*02:01-transgenic (ABabDII) mice [20], we succeeded in isolating high-affinity TCRs against self-antigens such as FLT3, against cancer testis antigens such as Mage-A1 and NY-ESO, and against viral antigens [21,22,23]. In these mice, mouse TCR α/β-gene loci are replaced with their human counterparts and mouse H2 molecules are exchanged for the human HLA-molecule HLA-A*02:01. The TCR repertoire of these mice has not been educated on normal human self-antigens and is therefore superior in generating and isolating TCRs for human self-antigens. The successful isolation of TCRs against viral antigens suggests that these mice might also be useful in generating neoantigen-specific T cells [24]. 

Here, we explored these different types of TCR repertoires to isolate specific TCRs directed against eight predicted HLA-A*02:01-restricted neoepitopes: patients’ TILs and PB lymphocytes (PBLs), PBLs from HLA-A*02:01-matched healthy donors, and T cells from primed HLA-A*02:01-transgenic ABabDII mice. We isolated six TCRs from four healthy donors directed against four neoepitope candidates, and one TCRs from ABaaDII repertoire directed against one neoepitope candidate (for which a healthy donor-derived TCR was also isolated). Our data indicate that successful isolation of neoepitope-specific TCRs depends, at least in part, on the TCR repertoire of the donor.

## 2. Materials and Methods

### 2.1. Patients and Patient Material

Fresh tumor tissue, PB, and clinicopathological data of patients with localized solid tumors and at high risk of relapse were collected during surgery at initial diagnosis within a research project that aimed to develop mutation-specific T cells for ATT (ethics committee ID EA1/265/14, Charité–Universitätsmedizin Berlin: Collaborative Research Grant Project “Targeting somatic mutations in human cancer by T cell receptor gene therapy”). Two HLA-A*02:01-positive patients, one with colorectal and one with ovarian cancer, were selected for detailed T cell repertoire analysis. 

### 2.2. Cell Cultures

All cell lines were cultured in RPMI 1640 medium (with stable Glutamine, Gibco), 10% FCS (Universität Heidelberg), 12.5 mM HEPES, 100 IU/mL penicillin, 100 µg/mL streptomycin (P/S), and 50 µg/mL gentamycin (complete medium, CM). Transduced PBLs were maintained in CM containing FCS purchased from PAN-Biotech (TCM). Cell line T2 and HEKT-GALV-g/p were kindly provided by the group of Prof. Uckert (Max Delbrück Center for Molecular Medicine, Berlin). Primary peripheral blood mononuclear cells (PBMCs) used for stimulations and expanded cytotoxic T lymphocytes (CTLs) were cultured in T cell medium containing 5% human serum type AB (hTCM, PAN-Biotec). Dendritic cells (DCs) were maintained in serum-reduced hTCM (1% human serum). Cytokine supplementation to the media is given in detail in the respective sections. All cytokines and growth factors, except for IL-2 (Proleukin^®^, Novartis, Basel, Switzerland), were purchased from PeproTech (Hamburg, Germany).

### 2.3. Identification of Neoepitope Candidates

Tumor and corresponding healthy tissue samples were analyzed by WES and RNA sequencing to identify expressed, non-synonymous somatic mutations. We have computed neoepitopes for all variants in protein-coding regions, that passed quality control filters, and variants showing at least one read from the mRNA sample (mRNA coverage, which covers the mutation locus (rna_coverage)). Additional filtration for higher mRNA level expression was not performed. The selection of candidate epitopes was based on high binding affinities to HLA-A*02:01 complex. Potential neoepitopes were predicted in silico for peptide-MHC (pMHC) binding using the artificial neural network algorithm NetMHCcon1.1a [25]. Neoepitopes were ranked according to their MHC-binding affinity. High-binding candidates were synthesized as short peptides (GenScript, Piscataway; NJ, USA) and encoded on tandem minigenes (TMG), ordered as gene-fragments (GeneArt, Thermo Fisher Scientific, Waltham, MA, USA), and cloned into expression vector pcDNA3.1(-). Neoepitopes encoded on TMGs were flanked by 10 cognate amino acids, and multiple predicted neoepitopes were each separated by an alanine spacer to ensure proper proteasomal cleavage of the epitope. Enhanced-green fluorescent protein (eGFP) served as a reporter and was separated from the upstream minigene sequences via a GSG-p2A element. Transcription of the TMG plasmid results in one single RNA transcript. Individual epitopes are separated by an AAY sequence linker that ensures proteasomal cleavage [26]. TMGs were used for in vitro transcribed RNA (IVT-RNA) synthesis followed by nucleofection of APCs or direct plasmid nucleofection of target cell lines.

### 2.4. TIL Cultures and TMG Stimulation

TILs were isolated by dissecting tumor tissues, obtained after resection of the primary tumor, into small fragments and culturing 1–2 fragments in hTCM supplemented with IL-2 (50 IU/mL), IL-7, and IL-15 (both 5 ng/mL) for 12 days. Half-medium change was done every 2 to 3 days with supplemented interleukins. CD8+ TILs were separated from residual tumor tissue after days 12 of TIL culture with MACS CD8+ T cell isolation kit (Miltenyi Biotec, Bergisch-Gladbach, Germany) and cryopreserved until use. For the initial stimulation T cells, autologous DCs were generated from plastic-adherent monocytes with GM-CSF (800 IU/mL) and IL-4 (1000 IU/m) in serum-reduced hTCM over 3 days according to Dauer et al. [27]. For stimulations of T cells with TMG-derived IVT-RNA, immature DCs were nucleofected using Human Dendritic Cell Nucleofector™ kit (Lonza Group AG, Basel, Switzerland) according to the manufacturer’s protocol. Cells were matured overnight with a cytokine-cocktail containing GM-CSF, IL-4, TNFa (10 ng/mL), IL-1ß (10 ng/mL), IL-6 (10 ng/mL), and PGE2 (1 µg/mL). Nucleofection efficiency was assessed by eGFP expression via flow cytometry 12 h after nucleofection. T cell stimulations were set up in wells of a 48-well plate, and 5 × 10^5^ T cells/well were co-cultured with nucleofected DCs (TMG-DCs) at an effector to target ratio (E:T) of 10:1, calculated according to the nucleofection efficiency. IL-7 and IL-15 (each 5 ng/mL) were added to the medium on day 3, and cells were fed on day 6, 8, 10, and 12. Cytokine concentration was doubled on day 8, and expanding CTLs were transferred to the next larger well plate if necessary. TILs were restimulated with TMG-DCs twice, as described above, followed by an overnight co-culture and staining for CD8α (SK1, APC-H7-conjugated, BD Biosciences), CD3 (SK7, PerCP-conjugated, BD Biosciences, Franklin Lakes, NJ, USA) and CD137 (4B41, PE-conjugated, BioLegend^®^, San Diego, CA, USA) for single cell FACS sorting and TCR sequencing.

### 2.5. Generation of Neoepitope-Specific T Cells from PB

Patient and healthy donor CD8+ T cells were negatively selected from PBMCs using MACS CD8+ T cell Isolation Kit and frozen until further use. CD8-depleted PBMC were used for DC generation and maturation, as described above, after plastic adhesion or frozen for the following peptide stimulations. The neoantigen-specific T cell culture was adapted from Wölfl et al. [28]. Briefly, CD8+ T cells were thawed and rested overnight in hTCM containing IL-7 (5 ng/mL). Naive CD8+ T cell fractions were obtained by depletion of CD45RO+ and CD56+ cells (MACS microbeads, Miltenyi Biotec, Bergisch-Gladbach, Germany). Non-naive cells were recovered from the column. Autologous mature DCs were individually loaded with peptides at 2.5 µg/mL for 2 h in reduced hTCM. Peptide-pulsed, washed DCs were then pooled and used for T cell stimulation with an E:T of 10:1 for each peptide-loaded DC faction in wells of a 48-well plate at 5 × 10^5^ cells/well. Expanding CTLs were maintained with IL-7/15 (5 ng/mL) as described above, and cytokine concentration was doubled on day 8 of co-culture. CTLs were restimulated once with irradiated, peptide-loaded, autologous, CD8-depleted PBMCs on day 14 of expansion culture. On day 10–12 after restimulation, peptide reactivity was screened for in an activation assay by measuring CD137 expression. Peptide-reactive CTLs were selected and sorted following a short co-culture with peptide-loaded T2 cells in the presence of monensin (GolgiPlug™, BD Biosciences, Franklin Lakes, NJ, USA) and an antibody against degranulation marker CD107a (H4A3, PE-conjugated, BioLegend^®,^ San Diego, CA, USA) for 3–4 h. Activated CD3+CD8+CD107+ CTLs were sorted for RNA isolation (RNeasy Micro Plus Kit, Qiagen, Hilden, Germany) into RLT buffer.

### 2.6. Immunization of ABabDII Mice

Animal experiments were performed according to institutional and national guidelines and regulations following approval by the responsible authority (Landesamt fuer Gesundheit und Soziales, Berlin, Germany). Mice were primed with synthesized predicted HLA-A*02:01-restricted peptides on day 0 with 150 µg of peptide in a 1:1 solution of Incomplete Freund’s Adjuvant (IFA, Sigma-Aldrich, St. Louis; MO, USA) and 50 µg CpG1826 (Novus Biologicals) by subcutaneous injection. Mice were boosted three times, the earliest on day 21 after priming. Blood was collected 7 days after each boost, and mouse PBMCs were cultured with 1 × 10^−6^ M peptide overnight for Interferon-γ (IFN-γ) detection. Mice with IFN-γ-secreting CD8+ T cells in the periphery were sacrificed, and spleen and inguinal lymph nodes were collected. Splenocytes were isolated, and CD4+ cells were depleted. Splenocytes were expanded for 10 days in RPMI 1640 with 10% FBS, HEPES, NEAA, sodium pyruvate, 50 µM β-mercaptoethanol, 20 IU/mL IL-2, and 10^−8^ M peptide. After expansion, peptide-reactive cells were sorted in RLT buffer for RNA isolation following IFN-γ secretion assay (Miltenyi Biotech, Bergisch-Glattbach, Germany) and staining with antibodies against mouse CD3 (clone 17A2, APC-conjugated, BD Biosciences, Franklin Lakes, NJ, USA) and mouse CD8 (53-6.7, PerCP-conjugated, BD Biosciences, Franklin Lakes, NJ, USA). 

### 2.7. TCRα/β Chain Identification

For single-cell (sc) TCR sequencing, expanded, single, viable CD3+CD8+ T cells were sorted into 96-well plates, and paired TCR sequencing was performed as previously described [29,30].

For bulk sorted CD3+CD8+CD107+ T cells, total RNA was extracted and 5′RACE-ready cDNA was synthesized using the SMARTer RACE kit (Takara Clonetech, Kusatsu, Japan). TCRα and β variable (TRAV/TRBV) chains were amplified as previously described [17] with TCRA (5′-CGGCCACTTTCAGGAGGAGGATTCGGACC-3′) or TCRB (TCRB: 5′-CCGTAGAACTGGACTTGACAGCGGAAGTGG-3′) gene-specific primers. Amplified PCR products were agarose-gel-purified, and corresponding bands were cloned using the Zero Blunt II TOPO vector (Invitrogen, Thermo Fisher Scientific, Waltham, MA, USA). A minimum of 20 bacterial clones were sequenced, and TRAV and TRBV chains were identified using the online IMGT.org vquest tool (http://www.imgt.org/IMGT_vquest/vquest, accessed between 24 February 2017 to 14 September 2021). The most frequent TRAV/TRBV chains were paired, and constant regions of identified TCRs were replaced by murine counterparts for expression in human PBLs. If more than one TRAV or TRBV sequence was dominant, all possible AV and BV chain combinations were assembled. TCRα and β chains were linked with a p2A element, and NotI and EcoRI restriction sites were added for restriction site cloning of the TCR cassettes into pMP71 γ-retroviral vector.

### 2.8. Virus Production and Transduction of PBLs

Packaging cell line HEKT-GALV-g/p, stably expressing MLV gag/pol and pALF-GALV, was transfected with 18 µg TCR-cassette-pMP71 vector to produce virus particle-containing culture supernatant. Culture supernatant was collected 48 h and 72 h after transfection. Healthy donor PBLs were activated on anti-hCD3/anti-hCD28 (clones HIT3a and CD28.2, respectively, BD Biosciences, Franklin Lakes, NJ, USA)-coated 24-well plates in 1 mL TCM supplemented with 400 IU/mL IL-2 at a density of 1 × 10^6^ cells/mL for 48 h before transduction. Activated PBLs were spinoculated (800× *g*, 90 min, and 32 °C) with 1 mL cell-free virus supernatant 48 h after activation. Protamine sulfate was added to the medium at 4 µg/mL to aid transduction. A second transduction was performed on the following day by spinoculating virus supernatant onto RetroNectin^®^ (Takara Clonetech, Kusatsu, Japan)-coated 24-well plates (900× *g*, 90 min, 4 °C) and transferring PBLs on coated wells. Transduction efficiency was assessed by flow-cytometry 72 h after the second transduction by staining PBLs against murine TCRβ constant chain (H57-597, PE-conjugated, BioLegend^®^, San Diego, Ca, USA), CD3, and CD8α. TCR-transduced (TCR-td) PBLs were further expanded in TCM, containing 400 IU/mL IL-2 and 5 ng/mL IL-7/IL-15, for one week. Transduced, expanded PBLs were either frozen after expansion phase or maintained in culture in TCM supplemented with 40 IU/mL IL-2 and 5 ng/mL IL-7/IL-15. 

### 2.9. Transfection of Cell Lines

Multiple myeloma cell line U266 (purchased from DSMZ-German Collection for Microorganisms) was transfected with neoepitope-encoding TMG-constructs in pcDNA3.1(+) plasmid by nucleofection with the Amaxa Nucleofector 4D device. One million cells were washed once with PBS, resuspended in 20 µL OptiMEM (Lonza Group AG, Basel, Switzerland), and transferred to the 4D Nucleofection strip. Two microgram plasmid DNA in 5 µL OptiMEM were added to the cell suspension, mixed, and nucleofected with the preset nucleofection program for cell line HL60. Viability and reporter expression were analyzed prior to coculture set-up by flow-cytometry.

### 2.10. CTL Screening and Functional Assays

CTLs were screened for peptide-reactivity after an overnight co-culture with peptide-loaded (1 × 10^−6^ M) target cells and staining for T cell activation marker CD137 by flow cytometry. Cells were stained against CD8α and CD137 (4B41, PE-conjugated, and BioLegend^®,^ San Diego, CA, USA). Peptide recognition of TCR-td PBLs was confirmed in co-culture assays of peptide-loaded target cells and PBLs in an overnight culture. Target cells were HLA-A2-positive cell lines T2 and U266. To assess the functional avidity of isolated TCRs, titration assays were performed on TAB-deficient T2 cells loaded with decreasing peptide concentrations. Secreted IFN-γ from cell-free supernatant was detected by ELISA (OptEIA™ sets, BD Biosciences, Franklin Lakes, NJ, USA) after an overnight co-culture. Tyrosinase peptide YMD (YMDGTMSQV) served as a peptide-loading control for T2 cells, and cells were pulsed at 1 × 10^6^ M and incubated with YMD-specific TCR T58-td PBLs. The mean functional avidity of TCRs was determined by measuring IFN-γ secretion to declining peptide concentrations of target cell, and the peptide dose at which the half-maximum of T cell activity was achieved (EC50) was determined.

For FACS analysis of T cell activation, cells were stained against CD8α, mTCRβ (PerCP-Cy5.5-conjugated, BioLegend^®^, San Diego, CA, USA), and CD137. For analysis of cytotoxicity, TCR-td PBLs and target cells were incubated for 3-4 h in the presence of monensin and anti-CD107α antibody followed by staining for CD8α and mTCRβ. CD107α-surface expression was assessed as a marker of degranulation via flow cytometry.

### 2.11. Statistical Analysis 

Data visualization and statistical analysis was performed using GraphPad Prism 7.0 (GraphPad Software, Version 7.00, GraphPad Software, La Jolla, CA, USA). EC50 values were calculated from peptide titration assays at the peptide dose at which the half-maximum of T cell activity was achieved. The data were normalized to the percentage of maximal (plateauing value) of response before curve fitting. Titration data were analyzed using 4PL model.

## 3. Results

### 3.1. Identification of Candidate Neoepitopes for an Ovarian Cancer Patient and Colon Cancer Patient 

WES and RNA sequencing were performed on tumor and healthy tissue for colon carcinoma patient BIH146 and patient BIH56 with advanced ovarian carcinoma. Patients’ characteristics are summarized in Appendix A. The WES of tissue samples was performed with an average sequencing depth of 681.6×/374.5× for BIH56 and 458.4×/234.4× for BIH146 and revealed 48 and 182 missense mutations for BIH56 and BIH146, respectively. Binding affinities of transcribed mutated peptide sequences for patient’s HLA alleles were predicted with NetMHCcons1.1a, and neoantigen candidates were selected according to their predicted MHC class I binding affinity (IC_50_). Identified somatic variants and a list of identified potential epitopes for HLA-A*02:01 are given in the Appendix A. For BIH146, seven HLA-A*02:01-restricted potential neoantigens were selected for further investigation, six of which had high (IC_50_ < 50 nM) and one of which had low (IC_50_ 653 nM) binding affinity. For BIH56, neoantigen candidates were restricted to five of the patient’s HLA-alleles, and IC_50_ ranged from 36 to 5720 nM with only one predicted candidate for HLA-A*02:01 (IC_50_ 63 nM). Selected neoantigen candidates are summarized in Table 1. For each patient, a TMG was constructed encoding for all selected candidates. TMG-constructs were used for IVT-RNA synthesis for stimulation of T cells, or for nucleofection of target cell lines to be used for testing endogenous epitope processing.

### 3.2. TCRs Identified from TIL or Patients’ PB Repertoire Did Not Show Reactivity against Selected Neoantigens

As a potential source of neoepitope-specific T cells, TILs and PB were analyzed in patient BIH146. The number of CD8+ cells isolated from TIL and PBMC was very limited for this study. We isolated a total of 1.15 × 10^7^ CD8+ T cells from TIL cultures of BIH146. Only 0.5–1 × 10^6^ CD8+ T cells from PBMCs for both patients could be used for each. Expanded CD8+ T cells, stimulated with autologous TMG-DCs, were sorted for single-cell TCR sequencing, and T cell clone frequencies were calculated where both α and β chains were detected. One dominant T cell clone was detected in stimulated TIL but not in stimulated patient-derived PBMCs. The identified TCR rearrangement 16A4 (Appendix A), expanded from a frequency of 2.3% (3/133 complete α/β rearrangements) before stimulation to 14.7% (10/68) after stimulation. Some expanded T cell clones were also present in mock-stimulated T cells and were thus not TMG-specific (Figure 1A). There was no clonal expansion in TMG-stimulated T cells derived from PB. Additionally, we stimulated TILs and PBMC-derived CD8+ cells with peptide-loaded APCs. Peptide stimulations of TILs (about 1 × 10^6^ cells per peptide) were carried out with single-peptide loaded APCs, which reduced the number of cells available for each individual peptide stimulation, compared to TMG stimulations, where all neoepitope candidates are encoded on one TMG-construct. Peptide stimulated T cells did not expand and could not be sorted for scTCR-sequencing. 

The identified expanded TCR 16A4 was cloned and expressed on donor PBLs. Following a co-culture with peptide-loaded target cells, recognition of neoantigen candidates by 16A4-td PBLs was assessed by measuring the expression of activation marker CD137. However, 16A4-td PBLs failed to recognize neoepitope candidates (Appendix A) when loaded onto target cells.

For BIH56, only patient-derived PBMCs but not TILs were available to isolate CD8+ T cells for stimulation with autologous peptide-loaded APCs. Expanded T cell clones were sorted and identified, and two TCRs were constructed (TCR 1C4 and 1A6, Appendix A). PBLs transduced with TCR 1C4 did not recognize peptide-loaded target cells, and TCR 1A6 showed unspecific activation by target cell line T2 independent (Appendix A) of peptide loading. 

In summary, although expansion of T cell clones from patients’ PBMCs and TILs could be detected, no neoepitope-reactive TCR was identified. An overview of all isolated TCRs, stimulation methods, and TCR repertoire used is given in Appendix A.

### 3.3. Detection of Neoantigen-Reactive CTLs after Stimulation of Healthy HLA-Matched Donor CD8+ T Cells

We used T cell repertoires of multiple healthy HLA-A*02:01-matched donors to stimulate neoantigen-specific T cells. HLA haplotypes of healthy donors are summarized in Appendix A. We were able to generate CTLs from a peptide pool of three predicted neoepitope candidates (CDH2_D660Y_, SLC35D1_T324S,_ and PIGM_V62L_), identified for patient BIH146, from the CD8+ T cells (bulk population containing naive and non-naive T cells) from donor A. Three expanded CTL lines specifically recognized target cells loaded with the peptide pool (Appendix A). 

Aiming to generate further CTL lines against neoepitope candidates of patient BIH146, T cell stimulation was repeated with the same donor as well as twice with four additional healthy donors. Here, we used a peptide pool of all seven predicted neoepitope candidates. Furthermore, to minimize the risk of unspecific T cell expansion, CD8+ lymphocytes were purified for naive T cells by sequential ‘untouched’ enrichment for CD8+ and naive cells. Both fractions, naive-enriched and non-naive CD8+ cells, were used for stimulations. Peptide recognition was assessed of expanded CTL lines in a screening assay measuring CD137 upregulation. A representative screening assay of all expanded CTL for donor A is shown in Figure 1B. Additional donors were treated accordingly (donors A-E). For patient BIH56, five HLA-A*02:01-matched donors (donors F-J) were stimulated with one predicted HLA-A*02:01-restricted neoepitope, SLC27A4_R408H_. Reactive CTL lines were then stimulated in a degranulation assay with the respective peptide, and cells were sorted for CD107a expression. Representative staining plots of CD8+CD107α+ CTLs are shown in Figure 1C for donors A and F. Expression of CD107α of reactive CTL lines are summarized in Figure 1D for all reactive donors. 

Overall, peptide-reactive CTLs could be generated from 4/5 donors for patient BIH146 directed against the seven predicted epitopes. For patient BIH56, 2/5 donors showed reactivity against the predicted neoepitope candidate. From one reactive donor, more than one CTL line could be generated, recognizing different neoantigen candidates. Interestingly, peptide CDH2_D660Y_ induced repeated and robust expansions of peptide-reactive CTLs in all reactive donors, whereas other peptides (e.g., MCOLN2_W390S_, TBC1D_E299K_, and CADPS2_E698Q_) induced expansion of specific CTL populations in only one donor. This suggests that the generation of peptide-specific CTLs is dependent on both the characteristics of the peptide and the donor repertoire.

### 3.4. Identified TCRs from the Human Repertoire Exhibit Reactivity against Neoantigen Candidates 

In order to identify the TCR sequences of peptide-reactive CTLs, TCRα and β chains of selected CD107α+ sorted CTL lines were identified. Most abundant TCRα/β chains were paired and cloned for expression in primary PBLs. We were able to identify highly prevalent TCRα/β variable regions for some, but not every, sorted CTL line. A summary of all identified TCRα/β rearrangements from a healthy donor repertoire, which were cloned and used for transduction of donor PBLs, is given Appendix A. TCR transduction efficiencies and IFNγ secretion of reactive TCRs after a co-culture with peptide-loaded target cells is shown in Figure 2A,B. TCRs 1628 and 6.2, isolated from donor A, were specific to predicted neoepitope candidate CDH2_D660Y_. Furthermore, TCR 56.11 and 56.48, isolated from two different donors, recognized the potential neoepitope candidate SLC27A4_R408H_, and TCR 3.6 and 5.1, isolated from two different donors, were specific to peptides MCOLN2_W390S_ and SLC35D1_T324S_, respectively.

In a peptide titration assay, EC_50_ values for TCR 1628 (5.4 × 10^−9^ M) and for TCR 6.2 (2.0 × 10^−7^ M) were determined. Peptide titration for TCRs 56.48 and 56.11 revealed -EC_50_ values of 1.1 × 10^−9^ M and 1 × 10^−6^ M, and EC_50_ values for TCR 3.6 and 5.1 were 4.9 × 10^−9^ M and 2.1 × 10^−9^, respectively, showing that we were able to identify high-avidity TCRs from the healthy human TCR repertoire (Figure 2C). 

Four reactive TCRs originated from naive-enriched CD8+ T cell fractions, and in the cases of TCR 1628 and 56.48, from the bulk CD8+ T cell fraction.

### 3.5. Immunization of ABabDII Transgenic Mice Led to the Isolation of a High-Affinity TCR 

ABabDII mice, transgenic for human TCRα/β gene loci and the HLA-A*02:01 molecule, were immunized four times with HLA-A2-restricted peptides, and three mice were immunized per peptide. Peptide-reactive T cells were sorted via IFN-γ capture assay from splenocyte cultures after reactivity was observed in PB. Most abundant TCRα and β sequences of reactive T cells were paired. The TCRα/β rearrangements are summarized in Appendix A. TCR mb11a4 showed only minimal recognition of its peptide (ATP11C_I1108R_) after incubation of TCR-td PBLs with peptide-loaded target cells at high peptide concentrations (≥10^−6^ M). We identified one high-affinityy TCR, TCR m875 (sorting plot Appendix A), and TCR-td PBLs recognized target cells loaded with peptide MCOLN2_W390S_ down to concentrations of 1 × 10^−11^ M with an EC_50_ of 6 × 10^−9^ M (Figure 2C). 

Taken together, likewise to the human repertoire, only a small fraction of the isolated TCRs from the mouse repertoire were specific to the predicted neoepitopes. One high-affinity TCR (m875) was specific against predicted neoepitope candidate MCOLN2_W390S_, for which one specific TCR (3.6) could also be generated from the human TCR repertoire.

### 3.6. Endogenous Processing of Predicted Neoepitope Candidates 

Multiple myeloma cell line U266, which naturally expresses HLA-A*02:01 and has been shown to be able to present internally processed, electroporated peptides [31], was nucleofected with TMG-constructs to confirm endogenous processing and presentation of candidate neoepitopes, for which specific TCRs were identified. Nucleofection efficiencies ranged from 14–45% (Appendix A). Figure 2D shows nucleofection efficiencies from TMG-nucleofected U266, which were used in a co-culture experiment with TCR-td PBLs (Figure 2E). TMG146-nucleofected U266 was recognized by PBLs expressing human-derived TCR 3.6 and murine-derived TCR m875, both specific to MCOLN2_W390S_. PBLs transduced with TCR 1628 and 6.2, however, did not recognize the nucleofected target cells, suggesting a lack of processing of the neoepitope candidate CDH2_D660Y_. The same is true for neoepitope candidate SLC35D1_T324S_ and TCR 5.1. Similarly, target cells nucleofected with TMG56, containing the HLA-A*02:01-restricted neoantigen candidate SLC27A4_R408H_, were not recognized by TCR 56.48-td and 56.11-td PBLs. Representative results from transduced PBLs of one donor are shown. Thus, endogenous processing of neoepitope candidates can be confirmed for 1/4 candidate epitopes, for which peptide-specific TCRs have been isolated.

## 4. Discussion

Adoptive transfer of neoantigen-specific T cells is considered one of the most attractive immunotherapeutic strategies. However, the optimal source of a TCR repertoire for their isolation is yet to be defined. Here, we exploited different types of TCR repertoires to isolate specific TCRs directed against predicted HLA-A*02:01-restricted neoepitopes from a colon and an ovarian cancer patient. Included TCR repertoires were the patients’ TIL and PB, the repertoire of different healthy HLA-A*02:01-matched donors, and the TCR repertoire of humanized AabDII mice. 

In total, we isolated six different TCRs from the healthy donor repertoire, which were specific to 4/8 predicted neoepitope candidates. The predicted neoepitope MCOLN2_W390S_ led to the generation of a specific TCR in both the human and the murine-derived TCR repertoire (TCRs 3.6 and m875). Additionally, endogenous processing was confirmed only for MCOLN2_W390S_. Some neoepitope candidates, for which we were able to isolate specific TCRs in one donor repertoire, did not result in the generation of dominant specific TCRs in other donor repertoires, indicating that the ability to generate specific TCRs is, in part, dependent on the donor TCR-repertoire. 

The patient’s own TCR repertoire would presumably be the ideal source for isolating individual neoantigen-specific TCRs. Tumor growth may already induce an immune response that leads to tumor-specific T cell expansion, at least in the immediate tumor environment. Furthermore, a successfully isolated TCR from the patient’s own repertoire would not raise safety concerns. However, we were not able to isolate a specific TCR for the predicted HLA-A*02:01-restricted neoepitopes from the patients’ own TCR repertoire. 

Limited success rates in detecting neoantigen-reactive T cells and isolating corresponding specific TCRs from patients’ repertoire have been observed before in several tumors, initially in melanoma [11]. This may be primarily due to the low frequency of neoantigen-specific T cells in TILs. Isolation of tumor-specific TCRs might be even more challenging in epithelial cancers, which frequently have immune-excluded or immune-desert phenotypes that hamper efficient T cell priming. Nonetheless, it has been shown that neoantigen-specific CD8+ and CD4+ T cells can indeed be isolated from TILs or from the PB T cell repertoires in metastatic epithelial cancers of some individual patients [32]. These tumor-specific T cells can be found exclusively in the memory T cell population. 

The limited number of T cells from TIL cultures in the patients analyzed in this study presented a major limitation considering the low frequency of neoantigen-specific T cells in TILs [11]. Infiltrating and peripheral neoantigen-specific T cell frequencies are estimated to be as low as 0.002% in cancer patients [7], which is up to two magnitudes lower compared to the EBV-specific T cell in seronegative donors (ranging from 0.3–0.8%), and almost four magnitudes lower compared to EBV seropositive donors, which range from 2.1 to 9.8% (with an average of 4.7%) [33]. Hence, it is desirable to expand specific T cells to higher frequencies or apply highly sensitive strategies for the detection and enrichment of neoepitope-specific T cells. In addition to using cultures of tumor fragments to enrich neoepitope-specific T cells, immediate identification of tumor-reactive T cells from the bulk TIL population after tumor resection is another way to isolate neoepitope-specific TCRs. This can be accomplished by sorting T cells expressing activation markers such as PD1, ideally combined with CD134/CD137, and followed by single-cell sequencing [34,35]. This approach would reduce the effort required to pair the a/ß TCR chains, as well as yield more sequencing depth. In addition, one would not risk losing tumor-specific T cells in the expansion cultures due to overgrowth of nonspecific T cell clones because exhausted tumor-specific T cells would not persist as well in culture. Regardless of which approach is used to isolate neoantigen-specific T cells, it would be desirable to collect tumor fragments from different locations from the resected tissue to account for tumor heterogeneity. Using the latter approach, TCR specificity would subsequently need to be determined using in silico predicted peptides. Moreover, it has been recently shown that most infiltrating lymphocyte clones, which are predominantly detected in the tumor tissue, are not necessarily tumor-specific [14]. 

Interestingly, most of the reported identified TCRs are not directed against HLA-A*02:01-restricted epitopes [3,16,36]. In another study, neoantigen-reactive T cell lines expanded for seven different HLA class I-restricted neoepitopes from PB of an ovarian cancer patient, but only one recognized an HLA-A*02:01-restricted neoepitope [37]. Since we have solely investigated the isolation of TCRs against potential HLA-A*02:01-restricted neoepitopes, the possibility remains that specific TCRs against neoepitopes, other than the here predicted ones, could potentially be isolated. Indeed, we have been successful in generating high-affinity TCRs specific to the HLA-B*07:02-restricted MyD88_L265P_ mutation [17]. 

In principle, it is possible to expand T cells against peptides of candidate neoepitopes from the naive T cell repertoire of cancer patients. However, limitations remain that influence the stimulation and priming success, namely, the low T cell precursor frequency and the often-limited amount of patient’s T cells available for neoantigen-specific in vitro priming, as was the case with the patient material in this study. To circumvent these limitations, we successfully used PB from HLA-A*02:01-matched healthy donors as a TCR repertoire source. Additionally, the use of HLA- and TCR repertoire-transgenic mice to isolate high-affinity TCRs, targeting tumor-associated or cancer germline antigens, circumvents mechanisms of tolerance and has allowed us to generate several useful TCRs [21,22,23] against self-antigen-specific TCRs. 

The limited success of inducing specific TCRs in transgenic mice against the here-predicted neoepitope candidates other than MCOLN2_W390S_ may be due to different factors, such as binding affinities of predicted epitopes to the human MHC-complex, or the immunogenic properties of the predicted candidates. Additionally, immunizing more animals (three mice per candidate were used) might have led to the generation of more specific TCRs. 

In general, the incidence of processed and immunogenic neoepitopes among predicted candidates is very low [38]. In this study, endogenous processing and the presentation of neoepitope candidates was confirmed for only one of the four predicted candidates, for which high-affinity TCRs were isolated. It cannot be excluded that neoepitope candidates with a predicted lower binding affinity, and thus not prioritized by our in-silico prediction approach, are possibly more immunogenic in vivo. A plethora of cumulative effects eventually determines which peptides function as immunogenic epitopes. This includes factors involved in antigen processing and presentation, which regulate the amount and quality of peptides presented on the cell surface, as well as factors that determine whether the presented peptide is recognized by T cells and is able to induce an immune response that renders the presented peptide immunogenic [39,40,41]. Besides inherent neoepitope properties, the individual donor TCR repertoire has a significant influence on the successful isolation of neoepitope-specific TCRs. TCR repertoires are not only affected by age and possibly the presence of acute or chronic infections but also by HLA polymorphisms [42,43]. The individual composition of HLA haplotypes of different donors might be another influencing factor considering the isolation success of neoantigen-specific TCRs. In this study, all stimulated donors were matched for HLA-A*02:01 only. It has been shown that TCR repertoire diversity is positively associated with polymorphisms in the HLA class I loci. While the effects of age and chronic viral infections negatively affect the TCR repertoire in general, HLA diversity may rather influence the ability to generate individual antigen-specific TCRs [42,44]. Recently, we have been able to isolate a specific high-affinity TCR for the recurrent MyD88_L265P_ mutation. In this case, we screened about 20 donors and were able to isolate 13 TCR sequences, from which seven high-affinity TCRs were identified from five donors [17]. Here, we tested five different donors for the potential neoepitopes of two patients. Including more donors might have led to the identification of more specific TCRs.

## 5. Conclusions

This study, as well as our experience with cancer-testis and other mutation-specific epitopes, suggest that, despite possible disadvantages in terms of safety, the generation of TCRs from allogeneic, partially matched healthy donors, or from humanized transgenic mice, might be a viable strategy for the development of mutation-specific ATT with defined specificity directed against cancer-specific mutations, especially in settings with limited patient material available. Here, we isolated specific TCR for the same neoepitope candidates from multiple healthy donors and the humanized mouse model. Our results also highlight the need for adequate predictions for neoepitope processing and immunogenicity.

Thus, to identify high-affinity TCRs, it might be necessary to screen multiple, up to 20, donor repertoires [17]. TCRs identified by this approach offer an ATT treatment option for patients with relapsed or refractory disease after revalidation of the presence of the identified neoepitope. Ideally, to prevent immune escape by loss of target antigen or HLA alleles, transfer of TCR-T cells directed against several neoantigens, restricted to different HLA alleles, would be beneficial.

## Figures and Tables

**Figure 1 cancers-14-01842-f001:**
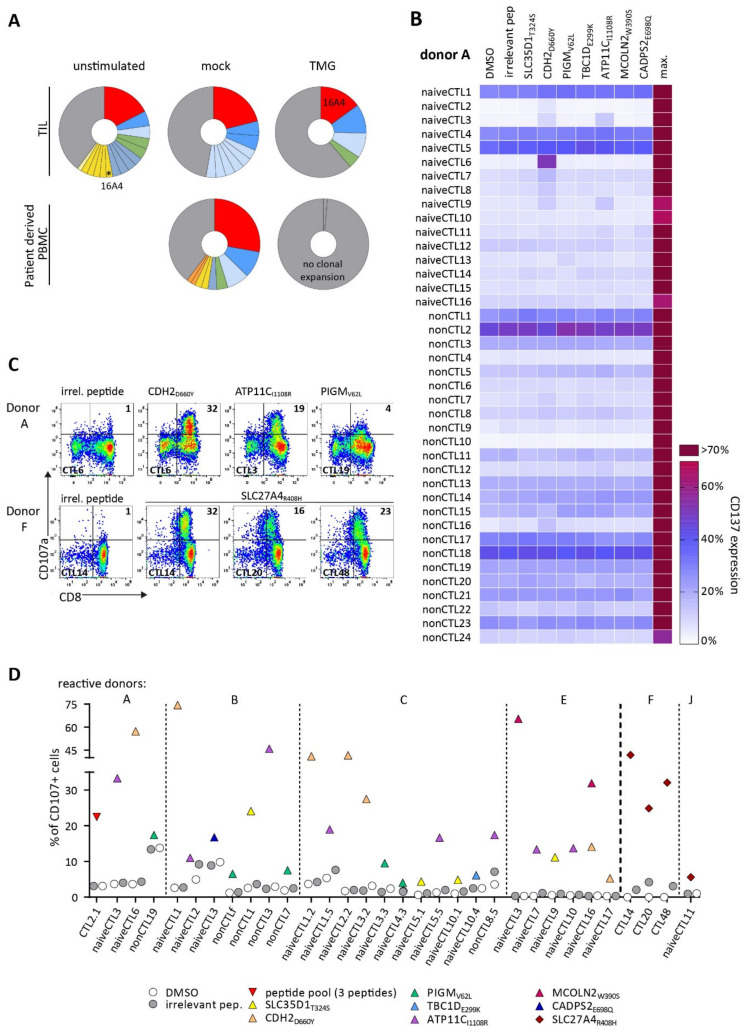
Stimulation and identification of patient- and donor-derived peptide-reactive T lymphocytes (CTLs). (**A**) Single-cell sequencing of TILs and PBMCs for patient BIH146. Pie charts show relative frequencies of detected TCRα/β sequences. The most frequent clone in TMG-stimulated TIL is indicated (16A4). Grey segments represent all TCRα/β rearrangements, which were detected only once per sample. (**B**) Peptide-expanded CTL lines were generated in a peptide stimulation containing all seven neoantigen candidates identified for patient BIH146. Each CTL line was screened for peptide recognition of individually loaded T2 cells, and T cell activation was measured by CD137 surface expression after 20 h. Representative results are shown for donor A-derived CTL lines, which were generated from naive-enriched (naiveCTL) and non-naive (nonCTL) CD8+ T cells. CD137 expression is depicted as a heatmap of the percentage of CD8+ cells. For each patient, five donors were stimulated twice. Max: maximal activation induced by PMA and Ionomycin activation. (**C**) CTL lines identified by screening assays were stimulated with their specific peptide. T2 cells were pulsed with peptide at 1 × 10^6^ µg/mL and used as target cells in a degranulation assay for 4h in the presence of monensin. Degranulation marker CD107a-positive cells were sorted. Representative FACS plots show CTL lines, which were generated from stimulations of naive and non-naive CD8+ T cells (donor A, upper panel) with a peptide pool of all seven peptides for BIH146 and bulk CD8+ T cells stimulated with one neoantigen candidate for BIH56 (lower panel, donor F). To estimate background activation, T2 cells were loaded with an irrelevant peptide and DMSO control. A representative control is shown (first row). (**D**) Summary of degranulation assays of all identified CTLs of all reactive donors (n = 6 reactive donors). Alive, single, CD3+CD8+ cells were gated for CD107a expression. Dotted lines separate donors, and the dashed line separates stimulations for patients BIH146 and BIH56.

**Figure 2 cancers-14-01842-f002:**
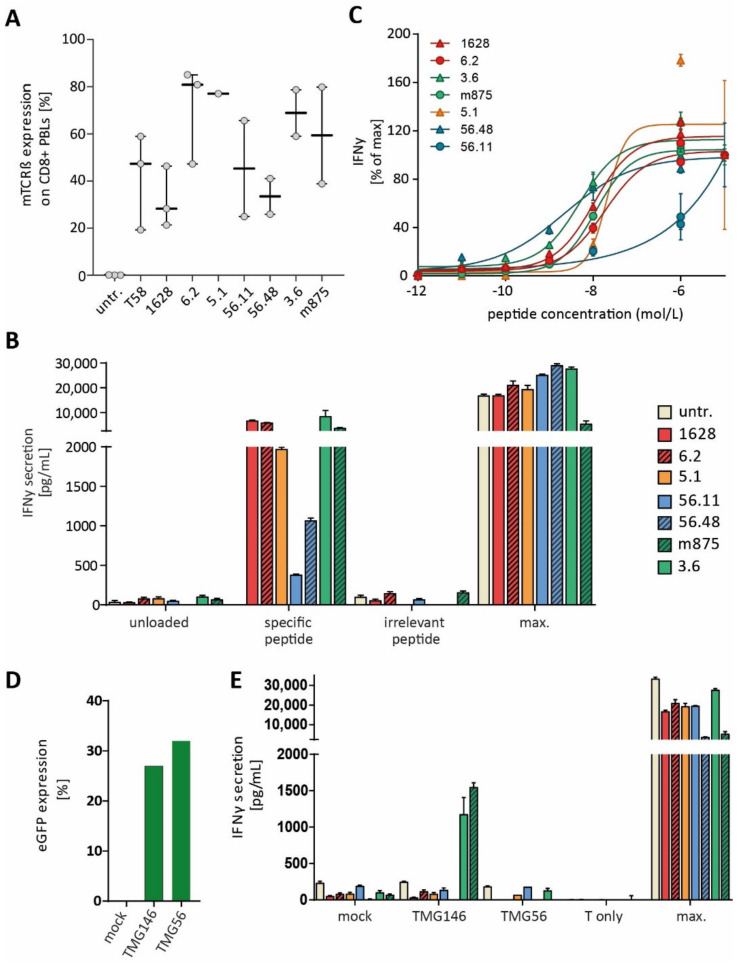
TCR specificity and endogenous processing of predicted neoepitope candidates. (**A**) TCR-cassettes were expressed in different donor PBLs (minimum n = 2), and expression of engineered TCRs was measured by staining for mouse TCRß constant region (mTCRß). Transduction efficiencies varied between 20 and 85% of CD8+ T cells. (**B**) The HLA-A2+ target cell line U266 was loaded with neoepitope candidates at 1 × 10^−6^ M and incubated with TCR-transduced (td) PBLs. Peptide-recognition was measured by IFNγ-secretion after overnight incubation. Representative data from one donor are shown, and samples were analyzed in duplicates. (**C**) Decreasing concentrations of peptides from 1 × 10^−5^–1 × 10^−12^ M were loaded onto T2 cells and co-cultured with TCR-transduced (TCR-td) PBLs. IFNγ secretion was measured from culture supernatant. (**D**) Nucleofection of tandem minigenes (TMG) encoding for neoepitope candidates into U266 cells. Nucleofection efficiency was assessed by eGFP-expression by flow cytometry. The graph shows nucleofection efficiency for the U266 cell, which was used for co-culture with TCR-transduced PBLs shown in graph (**E**). (**E**) Recognition of TMG-expressing U266 cells by TCR-td PBLs. IFN-y secretion was detected by ELISA after overnight incubation. Representative data are shown for one donor, and experiments were carried out in duplicates and for at least two different donors. Irrelevant peptide: non-specific or wildtype peptide; max: T cell activation cocktail, and untr.: untransduced. The same-colored bars/lines indicate specificity for the same neoepitope candidate; red: CDH2_D660Y_, yellow: SLC35D1_T324S_; blue: SLC27A4_R408H_, and green: MCOLN2_W390S_.

**Table 1 cancers-14-01842-t001:** Summary of neoantigens selected for TMG-construction for patients BIH146 and BIH56. Mutated amino acids of the respective epitopes are indicated in red. HLA-binding predictions were performed with NetMHC4.0; threshold rank for strong binders was 0.5, and for weak binders it was 2.00.

Patient	TMG Name	Epitope	Gene/ Mutation	HLA Restriction	% Rank	IC_50_ (nM)
146	TMG146	ILISYIGMV	SLC35D1_T324S_	A2:01	0.12	9.76
		RLNGYFAQL	CDH2_D660Y_	A2:01	0.6	44.92
		FLTEGERSPYL	PIGM_V62L_	A2:01	0.12	10.64
		MLFTIGQSKV	TBC1D1_E299K_	A2:01	0.6	43.62
		SLFPEILLRV	ATP11C_I1108R_	A2:01	0.15	12.01
		FLGTSTLLVSV	MCOLN2_W390S_	A2:01	0.5	36.71
		QLMEHSENGAV	CADPS2_E695Q_	A2:01	3.5	653.37
56	TMG56	HILSFVYPI	SLC27A4_R408H_	A2:01	0.8	62.41
		SPSRPPGPT	PLCB3_R918S_	B7:02	0.25	50.2
		LAVDTDEIEKY	NBPF_M674T_	B35:01	2.00	1356.1
		HSHELNGPY	CRYBB2_C38Y_	B35:01	0.2	36.43
		WLDGKHVVF	PPIAL4G_A128V_	C4:01	0.03	1679.01
		TKFDVQVLK	INF2_E670Q_	C4:01	0.4	4893.37
		YRQQAGRELL	IGSF9B_E63Q_	C7:02	0.01	48.03
		RRNPTGSVVM	PDXK_A305T_	C7:02	1.2	3589.15
		YRRDVHHVACY	KLHL22_Q281H_	C7:02	1.9	5719.99

## Data Availability

Generated data presented in this study on somatic variation and epitope predictions for the patients are available in the Appendix A. Further information on methods and data are available on request from the corresponding author.

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
