# Peer review of "Isolation of Neoantigen-Specific Human T Cell Receptors from Different Human and Murine Repertoires"

_cancers, 2022, doi:10.3390/cancers14071842_

Round 1
Reviewer 1 Report
This is a careful performed piece of work. I would like to see the following minor revisions:
- Please, describe clearly the advances/new knowledge this work is bringing to the field.
- Add the information on number of experiments performed, where relevant and the error bars, missing at least in the Supplemental Figure. Was there only one nucleofection (Fig. 2D)?
- Typos: Line 41, neoepitopes should be neoepitope candidates; Line 390, ELSA should be ELISA??
Reviewer 2 Report
See attachment.

Reviewer 3 Report
In the presented study, Corinna Grunert and coauthors investigated availability of possible T cell repertoire sources, such as PBLs from patients, healthy donors, and humanized mouse model, to isolate neoepitope-specific TCRs. Considering limited success so far by using patients’ own sources, this study is interesting and can provide valuable information and alternative strategy for isolation of the neoepitope-specific TCRs, by using T cell repertoires of healthy donor or transgenic mice of human TCR and HLA genes. Below I point out several minor issues to improve this manuscript.
Because many IDs were named for each material and sequence, it would be nice to show a summarized table (or figure) that can make linkages for neoepitope, TMG-construct, TCR ID, and T cell source.
For the selected neoantigen candidates (Table 1), how were their expression levels in the RNA-seq data? Their endogenous expression levels were taken into consideration for filtration of less reliable candidates?
Page 11, line 411-413; it seems that total 3 variants were investigated for the endogenous processing, therefore isn’t it 1/3 candidate epitope that could be confirmed finally?
Page 12, line 434-435; regarding the use of patient’s own T cell repertoire, it would be helpful to add discussion about any other feasible approach, for example, use of specific subpopulation of T cells (like CD8+PD-1+ subset).
The TCR repertoire diversity might be various individually, and thus be a critical reason to determine successful induction of neoepitope-specific T cells. Was there any difference in the TCR repertoire diversity among blood donors (as well as among transgenic mice)?
In Figure 2, there was no label for Figure 2B.
In the supplementary table 5 (uploaded Excel file), it was mis-labeled as “Table 4” and contained total 4 tables (Table 1-4).
